# Comparative Analysis of Predictive Interstitial Glucose Level Classification Models

**DOI:** 10.3390/s23198269

**Published:** 2023-10-06

**Authors:** Svjatoslavs Kistkins, Timurs Mihailovs, Sergejs Lobanovs, Valdis Pīrāgs, Harald Sourij, Othmar Moser, Dmitrijs Bļizņuks

**Affiliations:** 1Research Institute of Pauls Stradins Clinical University Hospital, LV-1002 Riga, Latvia; skistkin@gmail.com (S.K.); pirags@latnet.lv (V.P.); 2Institute of Smart Computing Technologies, Riga Technical University, LV-1048 Riga, Latvia; timur.mikhailov@yahoo.com (T.M.); dmitrijs.bliznuks@rtu.lv (D.B.); 3Interdisciplinary Metabolic Medicine Trials Unit, Division of Endocrinology and Diabetology, Medical University of Graz, 8010 Graz, Austria; ha.sourij@medunigraz.at; 4Division of Exercise Physiology and Metabolism, Institute of Sport Science, University of Bayreuth, 95447 Bayreuth, Germany; othmar.moser@uni-bayreuth.de; 5SIA “R4U”, LV-1016 Riga, Latvia

**Keywords:** diabetes, CGM, hypoglycemia, hyperglycemia, prediction, ARIMA, logistic regression, LSTM

## Abstract

Background: New methods of continuous glucose monitoring (CGM) provide real-time alerts for hypoglycemia, hyperglycemia, and rapid fluctuations of glucose levels, thereby improving glycemic control, which is especially crucial during meals and physical activity. However, complex CGM systems pose challenges for individuals with diabetes and healthcare professionals, particularly when interpreting rapid glucose level changes, dealing with sensor delays (approximately a 10 min difference between interstitial and plasma glucose readings), and addressing potential malfunctions. The development of advanced predictive glucose level classification models becomes imperative for optimizing insulin dosing and managing daily activities. Methods: The aim of this study was to investigate the efficacy of three different predictive models for the glucose level classification: (1) an autoregressive integrated moving average model (ARIMA), (2) logistic regression, and (3) long short-term memory networks (LSTM). The performance of these models was evaluated in predicting hypoglycemia (<70 mg/dL), euglycemia (70–180 mg/dL), and hyperglycemia (>180 mg/dL) classes 15 min and 1 h ahead. More specifically, the confusion matrices were obtained and metrics such as precision, recall, and accuracy were computed for each model at each predictive horizon. Results: As expected, ARIMA underperformed the other models in predicting hyper- and hypoglycemia classes for both the 15 min and 1 h horizons. For the 15 min forecast horizon, the performance of logistic regression was the highest of all the models for all glycemia classes, with recall rates of 96% for hyper, 91% for norm, and 98% for hypoglycemia. For the 1 h forecast horizon, the LSTM model turned out to be the best for hyper- and hypoglycemia classes, achieving recall values of 85% and 87% respectively. Conclusions: Our findings suggest that different models may have varying strengths and weaknesses in predicting glucose level classes, and the choice of model should be carefully considered based on the specific requirements and context of the clinical application. The logistic regression model proved to be more accurate for the next 15 min, particularly in predicting hypoglycemia. However, the LSTM model outperformed logistic regression in predicting glucose level class for the next hour. Future research could explore hybrid models or ensemble approaches that combine the strengths of multiple models to further enhance the accuracy and reliability of glucose predictions.

## 1. Introduction

Recently developed continuous glucose monitors (CGM) have made a revolution in the treatment and management of people with diabetes. Novel CGM offers real-time alerts for hypo- and hyperglycemia, as well as anticipated glycemic fluctuations, greatly enhancing glycemic control during meals and physical activity. However, due to the complexity of CGM systems, both individuals with T1D and healthcare professionals may face challenges when interpreting rapidly changing glucose levels caused by sensor delays (i.e., the inherent ~10 min discrepancy between interstitially measured and actual plasma glucose values) [1], and sensor malfunctions (e.g., compression hypoglycemia) [2,3].

Machine learning and neural networks have seen increasing use in predicting glucose fluctuations in people with diabetes when using CGM. The most widely used data for machine learning and glucose prediction models include glucose levels, insulin administration, carbohydrate intake, and physical activity/exercise. The prediction horizon, or the amount of time available to a diabetic patient before the predicted event occurs (hypoglycemia or hyperglycemia), should be sufficiently long to allow a timely response to prevent hypoglycemia or hyperglycemia. Recent advances in machine learning have led to the adoption of recurrent neural networks (RNNs) [4], including long short-term memory networks (LSTMs), as powerful tools for time series prediction. RNNs, particularly LSTMs, have proven effective in modeling sequences of data, making them well-suited for predicting future glucose levels based on historical measurements and other relevant features. Deep learning techniques like dilated recurrent neural networks (DRNN) have demonstrated significant potential in glucose forecasting, with a focus on capturing long-term dependencies [5]. Nevertheless, practical challenges remain, such as the need for extensive data and the potential for suboptimal performance due to varying lag lengths across individuals.

Scientists compared the effectiveness of multitask learning and sequential transfer learning, showing the superiority of multitask learning [6]. Additionally, researchers have explored interconnected lag fusion frameworks based on nested meta-learning for more accurate blood glucose forecasting [7]. While this approach appears promising, it can introduce complexity and computational overhead when applied to large datasets or individualized patient models. Bayesian regularized neural networks (BRNN) also offer comparatively good results [8]. Moreover, researchers are exploring innovative methods to improve CGM-based glucose prediction accuracy and reliability. These include techniques like Kalman smoothing [9] to correct inaccurate CGM readings caused by sensor errors, as well as ensemble empirical mode decomposition based on fractal dimension (FEEMD) and kernel extreme learning machine (KELM) models that incorporate novel signal processing and machine learning approaches to enhance prediction performance [10].

Certainly, it is crucial to recognize that there may be scenarios where nothing but the glucose level data is available, whether due to limitations in data collection or patient non-compliance with reporting additional metrics like physical activity or carbohydrate intake. In such cases, features extracted from the changes in glucose levels themselves can serve as valuable inputs for machine learning models. For instance, rate of change metrics, variability indices, and time-based features such as moving averages or seasonal decompositions can be engineered from the raw glucose data. These engineered features can add layers of nuance to the data, capturing the dynamism and trends in glucose fluctuations, thereby enabling more accurate and timely predictions. Features like rolling averages, standard deviations, or higher-order differences can help encapsulate the volatility or stability in the glucose levels, providing critical information even when other physiological metrics are missing. So, while the availability of a diverse feature set—including insulin levels, physical activity, and dietary intake—is ideal for predictive modeling, the absence of these does not preclude effective glucose level forecasting. By creatively utilizing features based on glucose level changes, we can still gain valuable insights into underlying physiological patterns and improve prediction accuracy. This approach ensures that machine learning models remain both robust and versatile in a variety of real-world conditions for effective diabetes management. Therefore, while the field has made considerable strides in leveraging complex machine-learning architectures for glucose prediction, the essence of effective modeling often lies in the nuanced selection of features. This is true whether one has access to a comprehensive set of variables like insulin administration and physical activity, or solely relies on glucose level data. In fact, when only glucose levels are available, features such as rate of change, variability, and time-based metrics can be engineered to improve the model’s prediction capabilities. With this comprehensive approach to feature selection in mind, the aim of this study was to investigate the efficacy of three different glucose predictive models—an autoregressive integrated moving average model (ARIMA), logistic regression, and long short-term memory networks (LSTM)—in predicting glucose levels after 15 min and one hour. This multi-model approach, coupled with an insightful feature set, aspires to present a robust framework for timely and accurate glucose level prediction, thereby contributing to the advancement of personalized diabetes management.

## 2. Materials and Methods

In this study, we compared and evaluated the performance of these models in predicting hypoglycemia, euglycemia, and hyperglycemia states. Hypoglycemia was defined as a glucose level below 70 mg/dL, while hyperglycemia was defined as a glucose level above 180 mg/dL. The study aimed to determine which model provided the most accurate and reliable predictions for glucose levels by assessing metrics such as precision, recall, F1-score, and accuracy.

### 2.1. Data Collection

The data used in this investigation came from two sources: a cohort study involving people with diabetes and simulation results obtained using the CGM Simulator [11]. The clinical cohort CGM data were acquired as part of the “Immune response to COVID-19 Vaccination in people with Diabetes Mellitus—COVAC-DM” study (EudraCT; Number 2021-001459-15), performed at the Interdisciplinary Metabolic Medicine Trials Unit at the Medical University of Graz, Austria. In this study 11 participants with type 1 diabetes used a CGM device prior to the first COVID-19 vaccination and data was collected through the second COVID-19 vaccination. In addition to sensor glucose readings, data on insulin dosing and carbohydrate intake were collected. The data has been published previously [12].

The CGM Simulator is a system designed for in silico testing of control algorithms, consisting of three principal components: (1) a large cohort of simulated “subjects” based on real individuals’ data and spanning the observed variability of key metabolic parameters in the general population of people with T1D; (2) a simulator of CGM sensor errors representative of three popular brands; and (3) a simulator of discrete subcutaneous insulin delivery via insulin pumps of two brands. For our research we utilized the software Simglucose v0.2.1 developed by Jinyu Xie [11]. The software is a freely available Python implementation of UVA/Padova T1D Simulator 2008 version. The generated in silico data covers ten virtual patients in each of three age groups {adults, adolescents, children} and spans 10 days. Each day, these virtual patients consumed three regular meals, with the option to take up to three additional snacks. Meal and snack sizes were randomized around pivot values. Additionally, the actual snack consumption probability was set to be 50% so that on average 1.5 snacks were consumed per day. Meal and snack times were also randomized around a fixed schedule.

The real patient plot shows glucose data (blue line) only, whereas the virtual patient plot exhibits glucose (blue and cyan lines) data as well as carbohydrate (red line) data. The cyan line represents a smoothed simulated trajectory of glucose level over which a sampling noise has been added to obtain the blue line. Please note the different scales for glucose (CGM, primary Y-axis) and carbohydrate (CHO, secondary Y-axis) levels.

The original dataset contained the following fields: patient ID, timestamp, CGM measurement, insulin dosage, and consumed carbohydrates. The raw real patient data was primarily recorded at 15 min time frequency, However, some minor frequency variability was present in the series and data gaps did occur. We pre-processed the raw data and brought it to 15 min frequency, taking care of gaps and any bad entries. The only pre-processing required for the in silico data was re-sampling it for 15 min frequency as the simulator generates it for each minute. Figure 1 above shows the three-day data excerpts for real and virtual patients. To develop our forecasting system, we used both virtual and real patient data. The forecasting results that we report are based on the real patient data only.

### 2.2. Feature Engineering

Feature engineering was at the core of our methodology, enabling us to derive factors used in forecasting from the original CGM time series. These additional features can be broadly classified into the groups listed in Table 1 below. Examples of the original CGM time series and some of its derived features are shown in Figure 2a,b below.

### 2.3. Forecasting Methodology

We limited the maximum forecast horizon to four steps, effectively representing 1 h. We compared three approaches to forecasting: an autoregressive integrated moving average model (ARIMA), multinomial logistic regression, and long short-term memory (LSTM) network. The ARIMA model utilized the CGM time series as input and produced forecasts of CGM differences over the next 15 and 60 min. These forecasts were then converted to CGM levels and transformed into glycemia classes using the formula shown in Table 1 above. On the other hand, the logistic regression and LSTM models used our engineered features and their lagged values (with lags up to 12) as input, with the glycemia class 15 and 60 min ahead as the target. We set feature parameters *M* and *N* in Table 1 to be 4, 8, or 12, i.e., multiples of 4 (which represents one hour in a 15 min time series). For each patient, we divided data into in-sample and out-of-sample sets. For each forecast horizon, the models were estimated in the sample and then tested out of the sample. To assess the robustness of this approach, we calculated in-sample and out-of-sample errors and confusion matrices for each patient and provided aggregated statistics across all patients.

The results of the glucose prediction system, employing ARIMA, logistic regression, and LSTM models, included out-of-sample aggregated confusion matrices and classification scores for two prediction horizons: 15 min and 1 h. These metrics were based on real patient out-of-sample data, providing an assessment of the accuracy and effectiveness of each model in predicting glucose levels. The duration of 15 min and 1 h were used for different tactics in managing hypoglycemia and hyperglycemia. Hypoglycemia can lead to immediate and severe consequences, including loss of consciousness and seizures. Predicting hypoglycemia 15 min in advance allows individuals with diabetes or healthcare professionals to take proactive measures to prevent a further drop in blood sugar. This may involve consuming a fast-acting carbohydrate. In contrast, a 1 h prediction might involve adjusting insulin dosage to prevent complications. Hyperglycemia can also have detrimental effects on health, particularly if sustained over an extended period. Predicting hyperglycemia 1 h in advance enables individuals with diabetes to adjust insulin, modify diet, or increase physical activity.

### 2.4. Population

The clinical cohort was part of the “Immune response to COVID-19 Vaccination in people with Diabetes Mellitus—COVAC-DM” study (EudraCT; Number 2021-001459-15), in which 11 subjects (5 males and 6 females) with type 1 diabetes used a CGM device prior to the first COVID-19 vaccination and data was collected through the second COVID-19 vaccination. The mean age of the participants was 47 +/−11 years, mean HbA1c of 57 +/−8 mmol/mol, and a total daily insulin dose of 38 +/−14 IU. Our previous study found that glucose variability was not impacted by vaccination in both type 1 and type 2 diabetes patients within the study cohort [12]. Consequently, CGM data can be considered reliable for further analyses.

## 3. Results

A comparative summary of recall and accuracy scores for our three models for the 15 min and 1 h forecast horizons is provided in Table 2. Detailed confusion matrices for each of the models and for each forecast horizon are listed in the subsections below. It is important to note that the confusion matrices are normalized over true labels, ensuring that the sum of values in each row equals 100%. Any minor discrepancies are cosmetic in nature and result from rounding.

### 3.1. ARIMA

The reported scores for the ARIMA model, with a prediction horizon of 15 min and 1 h, offer valuable insights into its performance in predicting glucose levels across different states.

For a 15 min forecast horizon, for hypoglycemia the ARIMA model demonstrated a recall of 0.60, indicating that the model successfully identified 60% of the actual cases of hypoglycemia. The recall for euglycemia was 0.87, indicating that the ARIMA model successfully captured 87% of the actual instances of euglycemia. For hyperglycemia the ARIMA model exhibited a recall of 0.87, indicating that the model successfully identified 87% of the actual instances of hyperglycemia. The overall accuracy of the ARIMA model for the 15 min prediction horizon was reported as 0.86, indicating that it correctly predicted the glucose levels for approximately 86% of the instances.

One-hour forecasts were of lower quality, with a significant drop for hypoglycemia (as clearly seen in Figure 3, 1-h) and a total accuracy of 0.63. This sort of reduction in forecast quality was rather expected given that for a longer forecast horizon the model resamples data at a lower frequency and the number of hypoglycemia observations was rather low compared to the other two classes.

### 3.2. Logistic Regression

The multinomial logistic regression showed the best results (out of the three models) in predicting the glycemic state for the first 15 min (Figure 4).

For hypoglycemia, the model demonstrated a recall of 0.98, meaning that the model successfully identified 98% of the actual instances of hypoglycemia. For euglycemia, the logistic regression model showed a recall of 0.91, indicating that the model successfully captured 91% of the actual instances of euglycemia. Regarding hyperglycemia, the logistic regression model exhibited a recall of 0.96, indicating that the model successfully identified 96% of the actual instances of hyperglycemia.

For the 1 h horizon the accuracy of the logistic regression model was reported as 0.69, indicating that it correctly predicts the glucose levels for approximately 69% of the instances.

Compared to the ARIMA model, the logistic regression demonstrated a more accurate prediction of the glycemia states effectiveness in distinguishing between different glucose levels.

### 3.3. LSTM

The LSTM showed the best results (out of the three models) predicting the glycemic state for one hour (Figure 5).

For hypoglycemia, the model demonstrates a recall of 0.87, meaning that the model successfully identified 87% of the actual instances of hypoglycemia. For euglycemia, the LSTM model showed a precision-recall of 0.63, indicating that the model successfully captured 63% of the actual instances of euglycemia. The recall for hyperglycemia was 0.85, indicating that the LSTM model successfully identified 85% of the actual instances of hyperglycemia.

The accuracy of the LSTM model predictions for the 1 h horizon was reported as 0.73, indicating that it correctly predicted the glucose levels for approximately 73% of the instances.

In summary, the LSTM model demonstrated reasonable precision, recall, and accuracy scores for predicting hypoglycemia, euglycemia, and hyperglycemia states. While the model’s performance after 15 min did not exceed the logistic regression model, it still showed increased effectiveness in distinguishing between different glucose levels and predicting the corresponding states at a 1 h horizon compared to the logistic regression.

## 4. Discussion

The results of this study shed light on the performance of three different predictive models, ARIMA, logistic regression, and LSTM, in predicting glucose levels after 15 min and one hour within various glycemic ranges.

Previous studies have compared the prediction accuracy of these models for glucose level prediction. For example, in a study by Sadegh Mirshekarian et al. (2019) [13], it was found that the LSTM model shows significantly better performance and results compared to ARIMA. Similarly, in another study by Mohebbi, Ali et al. (2020) [14] the LSTM model was determined to be the most effective for blood glucose prediction compared to ARIMA. In our study, both the logistic regression and LSTM models have exhibited the highest precision, recall, F1-score, and accuracy values compared to the ARIMA. However, the logistic regression model was more effective in accurately predicting hypoglycemia, euglycemia, and hyperglycemia states during the first 15 min, while LSTM excelled after an hour.

Looking ahead, a potential future solution could involve a hybrid predictive approach using different models. For example, a model created using two different approaches, such as logistic regression and random forest, accurately predicted hypoglycemic episodes with high sensitivities of around 95% and 94% and specificities of approximately 97% and 95% for prediction horizons of 0 to 15 and 15 to 30 min, respectively (Darpit Dave et al., 2021) [15]. In contrast, ARIMA provided a precision of 64% in hypoglycemia detection for prediction horizons of 30 min (Francesco Prendin et al., 2021) [16].

The results of our study provide valuable insights into the applicability of different predictive models, for individuals with irregular or hard-to-predict glucose fluctuations according to the situation. Currently, the 30 min interval is the most common in building a predictive model for glucose fluctuations [17]. There are also models that predict the level of fluctuations in glucose with a time interval of 60 min and 15 min. However, the question of the time needed to make decisions to prevent hypoglycemia or hyperglycemia remains open depending on various factors [18]. Blood glucose level prediction models have the potential to greatly improve the management and treatment of diabetes. By accurately forecasting future glucose levels, these models can help individuals with diabetes better adapt their insulin, diet, and physical activity to acutely avoid hypoglycemia and chronically improve the time in range (70–180 mg/dL). Among these models, the LSTM model, with its ability to capture complex patterns and dependencies in glucose readings, shows promise in identifying long-term (1-h) correlations between glucose levels and may potentially be used in the long-term prognosis related to exercise or specific food consumption. Furthermore, as future technologies advance and we move closer to fully closed-loop insulin delivery systems, it must be ensured that CGM systems are able to assess sensor glucose accurately with a minimum lag time, to avoid CGM-induced (fatal) dysglycemia.

For individuals currently using a regular insulin pump and insulin pen therapy, leveraging CGM-based glucose prediction can make more informed decisions regarding their diabetes management, including adjusting insulin doses or prospectively treating hypoglycemia. Additionally, the flexibility of neural networks allows them to incorporate various inputs, such as changes in diet or physical activity levels, to predict future glucose fluctuations. This versatility is particularly advantageous for individuals with unpredictable glucose patterns, as it enhances the accuracy and adaptability of the predictions. Maintaining blood glucose levels within the euglycemic range (70–180 mg/dL) is crucial for the prevention of acute and chronic complications, including severe hypoglycemia and the risk of micro- and macrovascular complications [19,20,21]. The inability to accurately determine the level of physical activity contrasts with the broad recommendations to increase physical activity in the context of metabolic and chronic disease. This disconnect between reported and directly measured physical activity is well known [22].

For these purposes, a better prediction model is needed to promote an optimized synergy between insulin dosing and daily activities as well as to handle sensor delay and bridge periods of sensor malfunction. Generally, the amplitude of glycemic excursions and risk of hypoglycemia strongly correlate with the intensity and duration of exercise as well as the duration of the fasting. Physical exercise is an important component in the management of type 1 [23] and type 2 diabetes due to its positive effects on diabetes management and the prevention of progression of macrovascular and microvascular complications [24], improved quality of life, and reduced prevalence of depression [25]. However, high-intensity exercise increases the risk of rapid deviations of glucose levels [26], including hypoglycemia [27]. Frequent and severe hypoglycemia is associated with early mortality [28], increased cardiovascular risk [29], reduced quality of life and fear of hypoglycemia [30,31], and reduced exercise efficiency, which discourages people with diabetes from being physically active [32,33]. The risk of hypoglycemia during and after exercise can be lowered if specific insulin-dose adjustments are made or by consuming additional carbohydrates [34]. Accurate real-time blood glucose prediction based on a variety of inputs, including short-acting insulin injections, has clinical implications and great potential for improving the quality of life and longevity of individuals with diabetes.

One of the benefits of using a neural network for this task is that it can handle a large amount of data and identify complex patterns that may be difficult for a human to discern. This can be especially useful for individuals who have irregular or hard-to-predict glucose fluctuations. However, it is important to acknowledge that neural networks are not perfect and there are some limitations to consider when using them to predict glucose fluctuations. For example, they can require a significant amount of data to make accurate predictions, which can be a challenge for people who do not consistently track their glucose levels or who have gaps in their data. Additionally, errors or biases in the data used to train the neural network can impact prediction accuracy. The use of machine learning has yielded encouraging glucose prediction accuracy results in relatively small study populations or in silico studies [34,35,36]. In some cases, these algorithms have combined two state-of-the-art models to calculate nutrition absorption and glycemia, while also evaluating insulin. Researchers even tried to combine the model with a personalized genetic algorithm, the Nelder–Mead method, and natural biorhythm [35].

Despite these limitations, neural networks show great promise as a tool for predicting and managing glucose fluctuations. By analyzing patterns in blood glucose readings, insulin doses, and other relevant data, a neural network can learn to make accurate predictions about how a person’s glucose levels will change over time.

## 5. Conclusions

The logistic regression model was more effective in accurately predicting hypoglycemia, euglycemia, and hyperglycemia states during the first 15 min. However, the LSTM model exceeded logistic regression prediction results during an hour prognosis. These findings suggest that different models may have varying strengths and weaknesses in predicting glucose levels, and the choice of model should be carefully considered based on the specific requirements and context of the application. Furthermore, future research could explore hybrid models or ensemble approaches that combine the strengths of multiple models to further improve the accuracy and reliability of glucose predictions. Additionally, the study could be extended by considering additional features or incorporating real-time data to enhance the predictive capabilities of the models.

The logistic regression model demonstrated superior accuracy in predicting hypoglycemia, euglycemia, and hyperglycemia states during the first 15 min. However, for longer-term predictions over one hour, the LSTM model outperformed logistic regression. These findings emphasize that different models exhibit varying strengths and weaknesses in predicting glucose levels, highlighting the importance of selecting the appropriate model based on the specific requirements and context of application.

## Figures and Tables

**Figure 1 sensors-23-08269-f001:**
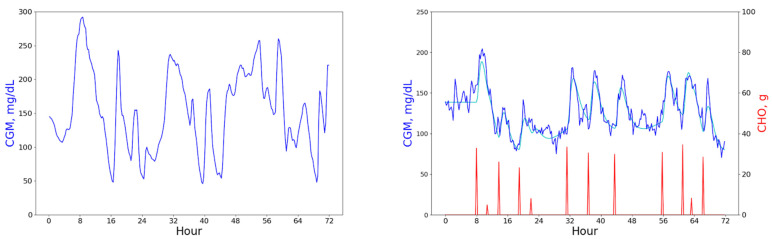
CGM data excerpts for real (**left**) and virtual patients (**right**).

**Figure 2 sensors-23-08269-f002:**
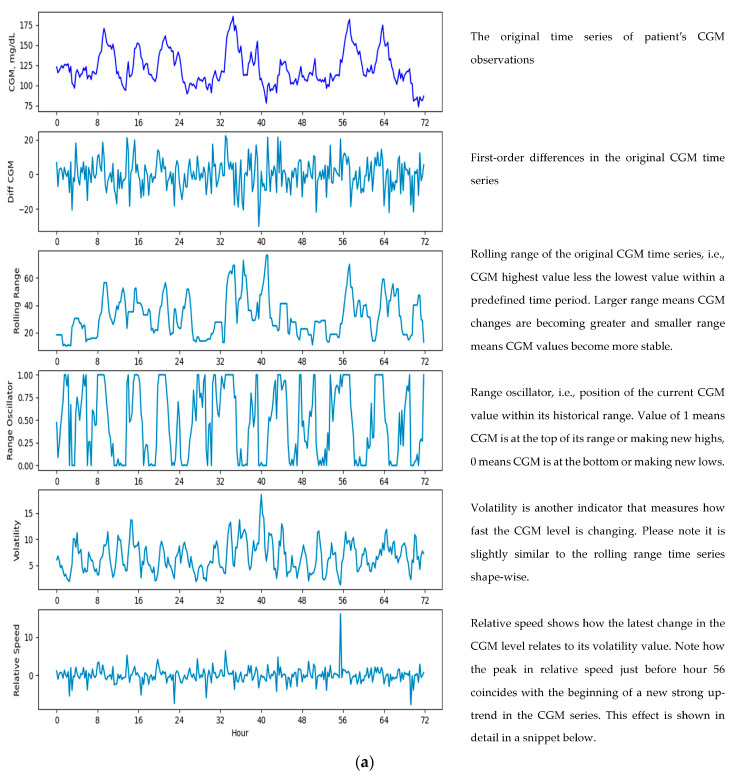
(**a**) Engineered features, (**b**) engineered features—a snippet for 54–62 h range.

**Figure 3 sensors-23-08269-f003:**
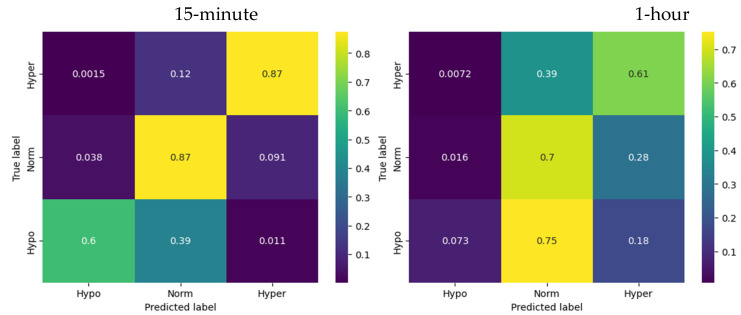
Confusion matrices for ARIMA model.

**Figure 4 sensors-23-08269-f004:**
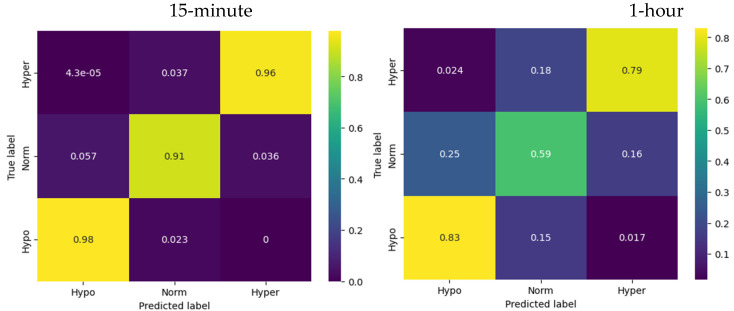
Confusion matrices for logistic regression model.

**Figure 5 sensors-23-08269-f005:**
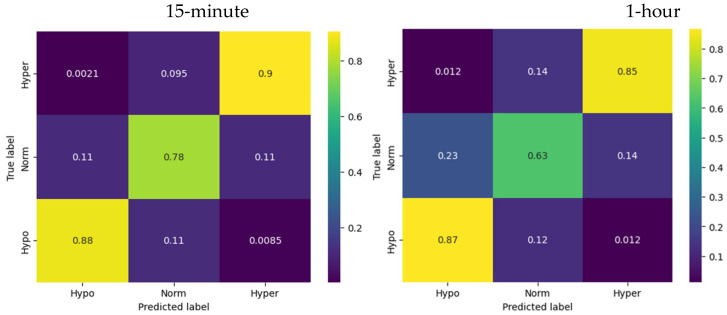
Confusion matrices for LSTM network model.

**Table 1 sensors-23-08269-t001:** Engineered features from the original CGM time series.

Feature Group	Formula	Meaning
CGM Differences	1st order: *d*CGM(*t*) = CGM(*t*) − CGM(*t* − 1) 2nd order: *d*^2^CGM(*t*) = *d*CGM(*t*) − *d*CGM(*t* − 1)	N-th order differences between consecutive CGM levels.
Rolling Range	RollingMIN(*t*, *N*) = min(CGM(*t*), CGM(*t* − 1)), …, CGM(*t* − *N* + 1) RollingMAX(*t*, *N*) = max(CGM(*t*), CGM(*t* − 1)), …, CGM(*t* − *N* + 1)	Maximum and minimum values calculated over a window of observations.
Range Oscillator	Oscillator(*t*) = (CGM(*t*) − RollingMIN(*t*, *N*))/ (RollingMAX(*t*, *N*) − RollingMIN(*t*, *N*))	Position of the latest value in the rolling range. Range Oscillator value of 1 means it is at the top of the range, 0 means it is at the bottom, and a value in between indicates its relative position.
Volatility	MAD(*t*, *M*) = sum(abs(*d*CGM(*t*)), abs(*d*CGM(*t* − 1)), …, abs(*dCGM*(*t* − *M* + 1))/*M*	Effectively represents the variability of the underlying data. Can be measured by rolling standard deviation of the CGM differences, mean absolute deviation (MAD), etc.
Relative Speed	RS(*t*, *M*) = *d*CGM(*t*)/Volatility(*t* − 1, *M*)	Ratio of level change value to volatility, which effectively means how fast the latest change is compared to the rolling metric. Positive relative speed means the underlying time series is rising and negative relative speed means its falling. Relative speed values outside the range [−1, +1] can indicate trend acceleration, and inside the range—its deceleration.
Glycemia Class	+1 (hyper) if CGM(*t*) > 180 GC(*t*) = 0 (norm) if 70 ≤ CGM(*t*) ≤ 180 − 1 (hypo) if CGM(*t*) < 70	This is effectively what we aim to predict—hypoglycemia, hyperglycemia or normal

**Table 2 sensors-23-08269-t002:** Recall and accuracy scores for ARIMA, logistic regression and LSTM models.

	ARIMA		Logistic Regression		LSTM	
	**15-min**	**1-h**	**15-min**	**1-h**	**15-min**	**1-h**
Glycemia state						
Hyper	0.87	0.61	0.96	0.79	0.90	0.85
Norm	0.87	0.70	0.91	0.59	0.78	0.63
Hypo	0.60	0.07	0.98	0.83	0.88	0.87
Accuracy	0.86	0.63	0.93	0.69	0.84	0.73

## Data Availability

The datasets used and/or analyzed during the current study are available from the corresponding author on reasonable request.

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
