# Peer review of "Comparative Analysis of Predictive Interstitial Glucose Level Classification Models"

_sensors, 2023, doi:10.3390/s23198269_

Round 1
Reviewer 1 Report
1. The manuscript can be titled more appropriately. Please note you are doing in-advance glycaemic classification and not glucose level prediction which the current title represents. I suggest a title like:
Comparative Analyses of Interstitial Models for Predictive Glycemic Classification in Type 1 Diabetes
2. Authors must be able to support the importance of performing such comparative analysis in the argument paragraph of the Introduction. I understand the importance of the topic in general but solely comparing three widely used models could imply a lack of sufficient novelty in the work. Where do your novelty and contributions lay?
3. The introduction is extremely weak and insufficient. I suggest the author expand it significantly by including more background and surveying recent studies such as:
· Multitask Learning Approach to Personalized Blood Glucose Prediction. (IEEE J. Biomed. Health Inform)
· Dilated Recurrent Neural Networks for Glucose Forecasting in Type 1 Diabetes. (J. Healthc. Inform. Res)
· Blood Glucose Prediction with Variance Estimation Using Recurrent Neural Networks. (J. Healthc. Inform. Res.)
· Blood Glucose Level Time Series Forecasting: Nested Deep Ensemble Learning Lag Fusion. (Bioengineering)
· Novel Blood Glucose Time Series Prediction Framework Based on a Novel Signal Decomposition Method. (Chaos Solitons Fractals)
· A Comparison of Different Models of Glycemia Dynamics for Improved Type 1 Diabetes Mellitus Management with Advanced Intelligent Analysis in an Internet of Things Context. (Appl. Sci.)
4. Figure 1 does not look high quality to me.
5. Captions of the tables and figures are not informative enough. Also, consider table notes or figure notes for listing abbreviations and extra information needed for further clarity. Note that tables and figures must be self-representative.
6. Strongly recommend that Table 1 be removed and information be given in the form of text.
7. Maybe use “a snippet” instead of “a zoomed version”.
8. The three confusion matrixes can be combined. This way we have all the results in one place and this facilitates side-by-side analysis.
9. I could not understand why Table 2 includes only results for the ARIMA model.
see comments file for authors
Reviewer 2 Report
Data from “The same pattern observed in hyperglycemia - ARIMA model (60%, 1 hour), logistic regression (96%, 15 minutes) and LSTM (85%, 1 hour)” in the abstract are not compared in the same duration.
Introduction lacks the recent advancements in CGM prediction, including different machine learning algorithms and their outcomes. In addition, deep learning technologies have been widely used in pattern recognition, object detection and prediction. As a result, this section must be written by including more state-of-the-art techniques.
Blood samples of 11 participants was collected through the second COVID-19 vaccination (Line 82 -84). People with chronic diseases are more vulnerable to COVID-19 viruses. Data acquired after the vaccination may not be suitable for further analysis due to the possible impact of the changing immune system. Authors must provide proof on data reliability.
Figure legends or running notes must be provided for curves overlapped in the same graph (e.g. Figure 1). Equations and parameter selection (Table 1) criteria should be scientifically described instead of using simple words or phrases, e.g. what is the size of observation window for rolling range; exact definition of mean absolute deviation?
Results of confusion matrices are too difficult to understand because the true positive rate is more that 100% in some circumstances (Figure 4 and Figure 5).
Typos and grammar mistakes can easily be spotted and language quality must be enhanced with the help of professional native speakers.
Round 2
Reviewer 1 Report
No additional suggestions. Thank you for considering the concerns
Author Response
We sincerely appreciate your valuable feedback. Thank you very much!
Reviewer 2 Report
Experimental results must be carefully checked, e.g. there exists an obvious mistake in the abstract "the LSTM model turned out to be the best for hyper- and hypoglycemia classes, achieving recall values of 85% and 85% respectively."
In the main text, recall rate for hypoglycemia classification is 87%
Revised version is much better than the origin except for some minor mistakes and typos.
Author Response
We sincerely appreciate your valuable feedback. We have carefully addressed the error in the abstract (line 30) and have diligently rectified minor language errors in English.